# Segmental Pulse Volume Recordings at the Forefoot Level as a Valuable Diagnostic Tool for Detection of Peripheral Arterial Disease in the Diabetic Foot Syndrome

**DOI:** 10.3390/biomedicines13061281

**Published:** 2025-05-23

**Authors:** Andreas Nützel, Lilly Juliane Undine Reik, Maximilian Hamberger, Christian Lottspeich, Sinan Deniz, Anja Löw, Holger Schneider, Hans Polzer, Sebastian Baumbach, Michael Czihal

**Affiliations:** 1Division of Vascular Medicine, Department of Medicine IV, LMU University Hospital, LMU Munich, 80336 München, Germany; andreas.kmnuetzel@med.uni-muenchen.de (A.N.); lilly.reik@med.uni-muenchen.de (L.J.U.R.); holger.schneider@med.uni-muenchen.de (H.S.); 2Department of Orthopaedics and Trauma Surgery, Musculoskeletal Center Munich, LMU University Hospital, LMU Munich, 80336 München, Germany; 3Division of Endocrinology, Department of Medicine IV, LMU University Hospital, LMU Munich, 80336 München, Germany; christian.lottspeich@med.uni-muenchen.de; 4Department of Radiology, LMU University Hospital, LMU Munich, 80336 München, Germany; sinan.deniz@med.uni-muenchen.de

**Keywords:** diabetic foot syndrome, peripheral arterial disease, medial arterial calcification, pulse volume recordings, upstroke time, maximum systolic amplitude, systolic ankle pressures, ankle brachial index, angiography

## Abstract

**Introduction:** Evidence for the diagnostic yield of noninvasive diagnostic assessment for the diagnosis of peripheral arterial disease (PAD) in diabetic foot syndrome (DFS) is poor. Pulse volume recordings (PVRs) at the forefoot level could be a valuable diagnostic tool in the presence of medial arterial calcification. **Patients and methods:** Patients with DFS who underwent invasive angiography between 01/2020 and 11/2024 and had corresponding PVRs performed within 30 days prior to the procedure were included. DFS was classified according to the Wagner–Armstrong classification. Clinical characteristics and hemodynamic parameters, including systolic ankle pressures and ankle–brachial index were recorded. PVRs were analyzed semiquantitatively by investigators blinded to the clinical information and quantitatively with determination of upstroke time (UST), upstroke ratio (USR), and maximum systolic amplitude (MSA). Angiographic PAD severity was graded according to the GLASS classification. Statistical analysis included univariate significance tests, 2 × 2 contingency tables, receiver–operator characteristic (ROC) analysis and determination of interobserver agreement. **Results:** In this study, 90 extremities of 70 patients were analyzed, 47 of whom had an ABI ≥ 1.3. Critical limb-threatening ischemia with non-pulsatile PVRs was evident in 6.7%. An abnormal PVR curve morphology (mildly or severely abnormal) yielded a sensitivity and specificity of 63.3% and 85.7% for detection of severe PAD (GLASS stages 2 and 3). Interobserver agreement of semiquantitative PVR rating was substantial (Cohen’s kappa 0.8) in 51 evaluated cases. For detection of any PAD (GLASS ≥ 1) or severe PAD (GLASS ≥ 2), we found the highest diagnostic accuracy for MSA (area under the curve [AUC] 0.89 and 0.82). With a cut-off value of 0.58 mmHg, MSA had a sensitivity of 91.4% and a specificity of 80.8% for detection of any PAD (GLASS ≥ 1). MSA with a cut-off of 0.27 mmHg had a sensitivity of 72.2% and a specificity of 77.1% for detection of severe PAD, whereas the sensitivity and specificity for detection of inframalleolar disease were 62.9% and 69.4%, respectively. Results were consistent in subgroup analyses. **Conclusions:** PVRs with extraction of quantitative features offer promising diagnostic yield for detection of PAD in the setting of DFS. MSA outperformed UST and USR but showed limited capability of detecting impaired inframalleolar outflow.

## 1. Introduction

The global disease burden of diabetes mellitus has incrementally risen in past decades, with the disease affecting an estimated 828 million people worldwide in 2022 [1]. Diabetic foot syndrome (DFS) is an important, frequent complication of diabetes mellitus, affecting approximately 18.6 million persons living with diabetes mellitus each year. The individual and socioeconomic consequences are enormous, as affected patients face a substantial risk of minor or major amputation (up to 20% of affected individuals), loss of mobility and quality of life, and increased mortality, mainly due to cardiovascular events [2].

While diabetic neuropathy along with abnormal pressure load of the foot and the toes are the main risk factors for development of DFS, and bacterial infection is the major complicating factor predisposing to limb loss, peripheral arterial disease (PAD) resulting in impaired arterial foot perfusion negatively impacts wound healing in DFS [3]. Current guidelines postulate “straight-line inflow to the foot” as a “conditio sine qua non” for sufficient wound healing and advocate aggressive, primarily endovascular revascularization strategies to achieve limb salvage [4]. However, noninvasive diagnosis of PAD, which commonly affects the cruropedal arteries in diabetic patients, is impaired secondary to medial arterial calcification (MAC). Specifically, physical examination, systolic ankle pressure measurement, and color duplex sonography of the below-the-knee arteries are hampered due to the circumferential calcification resulting in loss of compressibility of the affected arteries [2,4].

Several alternative noninvasive diagnostic measures, including systolic toe pressure measurement and transcutaneous oxygen measurement, have been proposed as useful in the diagnosis of PAD in patients with MAC [5,6]. Pulse volume recordings (PVRs) are routinely performed in daily vascular medicine practice in the diagnostic workup of suspected PAD, are easily obtained by assistant staff, and semiquantitative analysis is quite simple. Among other tests, PVRs have been recommended in guidelines on noninvasive vascular testing for diagnosis of critical limb-threatening ischemia [4,7]. The widely used Rutherford classification system incorporates flattened or barely pulsatile PVRs at the ankle or metatarsal level as an objective diagnostic criterion of CLTI [8]. However, the 2024 intersocietal guidelines on the management of the diabetic foot do not even mention PVRs as a potential diagnostic method, and recent systematic reviews did not include any study testing PVRs for detection of PAD in diabetes mellitus and wound healing prediction in DFS [5,6,9]. This is due to the fact that the evidence for the diagnostic accuracy of PVRs in diabetic and non-diabetic patients with MAC is poor, and most of the available studies were performed with outdated technology.

Assessment of quantitative parameters of PVRs may offer an additional diagnostic benefit, particularly in patients with impaired but not critically reduced arterial foot perfusion, but this diagnostic approach has not been tested so far in comparison to invasive angiography. We aimed to analyze the diagnostic yield of the forefoot of PVRs with both semiquantitative and quantitative methods for the detection of PAD of limbs affected by DFS.

## 2. Patients and Methods

### 2.1. Inclusion and Exclusion Criteria

We identified all patients who underwent digital subtraction angiography of the lower extremity arteries with complete visualization of the crural and pedal arteries between 01/2020 and 11/2024. Patients with an established diagnosis of diabetes mellitus suffering from nontraumatic foot wounds were included in this study, provided that PVRs of the forefoot had been recorded within 30 days prior to invasive angiography of the arteries of the affected leg(s). Several exclusion criteria were applied, as listed in Table 1. A flowchart illustrating the selection of eligible patients is given in Figure 1. Due to the retrospective study design, we did not calculate the sample size.

### 2.2. Clinical Characterization

Included patients were characterized with respect to medical history, including type of diabetes mellitus, preexistent PAD, previous revascularization and/or amputation, and current medication. The diagnosis of diabetes mellitus was considered to be confirmed when, according to the available medical documentation, the diagnostic criteria of the American Diabetes Association were met prior to the occurrence of the nontraumatic foot wound (fasting plasma glucose ≥ 126 mg/dL (7.0 mmol/L) after at least 8 h of fasting; plasma glucose ≥ 200 mg/dL (11.1 mmol/L) at 2 h after drinking a glucose solution; HbA1c > 6.5%. The type of diabetes mellitus was categorized as type 1, type 2, and other specific types, as suggested by the ADA [10].

The location and severity of the DFS were classified according to the Wagner–Armstrong classification, with the Wagner grade (numbers from 0 to 5) describing the depth and extension of the foot wound and the Armstrong stage (letters from A to D) reflecting the clinical impression of the presence of infection and/or ischemia [11,12]. Furthermore, foot wounds were categorized according to the SINBAD classification, as proposed by the international working group on the diabetic foot [5]. After cultivation of wound swabs on nutrient media, the bacterial isolates were characterized using microscopy (Gram staining), mass spectrometry, and polymerase chain reaction, as required.

Vascular physical examination findings and systolic ankle pressures/ankle brachial index were recorded, as well as laboratory findings, including blood count, C-reactive protein, HbA1c, and estimated glomerular filtration rate. Determination and calculation of the ABI followed broadly accepted recommendations [13]. Critical limb-threatening ischemia (CLTI) was defined according to the Rutherford classification [8].

### 2.3. Segmental Pulse Volume Recordings

All PVRs were performed using the AngE^TM^ system (SOT Medical Systems, Maria Rain, Austria). In order to recognize volume changes induced by pulsatile blood inflow into the legs, pairs of pneumatic measuring cuffs were inflated at six positions on the lower extremities (bilateral thighs, calves, and forefeet) to predefined pressures, which then were kept constant by means of two-way valves. The standard pressure applied to the forefoot cuffs was 50 mmHg. The pressure curves resulting from volume changes were recorded using pressure sensors with a time resolution of 1 ms and an amplitude resolution of 18 bits. The derived pressure signal was digitized by the pressure transducer and transmitted directly to the microcontroller, thus enabling an almost distortion-free display of the pulsations. The pressure signals recorded by the microcontroller were transmitted in packets to the computer via the USB interface and processed and graphically displayed by the associated AngioExperience^TM^ software. In order to ensure a meaningful representation of the signals, the absolute cuff pressure was not displayed, and only the pulsations transmitted from the vessels to the cuffs were shown graphically.

PVRs were assessed semiquantitatively with regard to curve morphology (side and level differences in amplitude, systolic upstroke, and diastolic dicrotic notch) (Figure 2A). A normal forefoot PVR was defined by a sharp upstroke, a scooped or flat interval between peaks, and a dicrotic notch at the diastolic descending part of the curve. PVR was classified as mildly abnormal when the dicrotic notch and the flat interval between curves were lost, while the upstroke remained steep. PVR was classified as severely abnormal when, in addition to the above-mentioned changes, the amplitude was flattened and the upstroke became prolonged (equal upstroke and downslope period). PVR was classified as non-pulsatile in the absence of visible pulsations, corresponding to CLTI [14].

For the purpose of this study, stored PVRs at the forefoot level were further assessed quantitatively for total wavelength (in ms), upstroke time (UST, in ms), upstroke ratio (USR, upstroke time divided by total wavelength), and maximum systolic amplitude (MSA, in mmHg) (Figure 2B) with the AngioExperience^TM^ software. In patients who underwent repeated PVRs after revascularization, we assessed sensitivity to change in the semiquantitative and quantitative measures.

### 2.4. Digital Subtraction Angiography

Digital subtraction angiographies were performed on an Artis Zee system (Siemens Healthineers, Forchheim, Germany). Digitally stored films of diagnostic angiography series were reviewed by an experienced radiologist blinded to the clinical and hemodynamic information. As patients with obstructions of the iliac arteries were excluded from this study, analysis comprised the femoropopliteal axis and the below-the-knee/below-the-ankle arteries. Severity of arterial obstructions was assessed by the GLASS classification, as proposed by the Global Vascular Guidelines on CLTI [4]. Briefly, lower extremity arterial lesions below the inguinal ligament were categorized according to severity (non-stenotic lesions, stenosis, and occlusion) and lesion length. The femoropopliteal axis and the crural arteries were judged separately. For the purposes of this study, all below-the-knee arteries were scored separately. The artery (preferentially a tibial artery) that contributed most to foot perfusion (target artery path) was taken into account for calculation of the overall GLASS score. In addition, the below-the-ankle outflow was scored as follows (with inframalleolar descriptors): P0, at least one patent foot-supplying artery with patent pedal arch; P1, severely diseased or absent pedal arch; and P2, no artery crossing the ankle towards the foot. Based on the three arterial levels, a global staging of disease severity was determined (grades 0-III).

### 2.5. Statistical Analysis

Clinical cohort characteristics were analyzed descriptively. Receiver operating characteristics analysis was applied for assessment of the diagnostic accuracy of quantitative PVR parameters for the detection of significant PAD. Diagnostic accuracy (sensitivity, specificity, and positive and negative predictive values) of the calculated cut-off values and of semiquantitative PVR ratings for detection of significant PAD were assessed using 2  ×  2 contingency tables, with subgroup analyses for patients with and without incompressible ankle arteries as shown by elevated ABI values ≥ 1.3 and for patients with foot infection. Univariate group comparisons were performed using χ^2^-test (categorical variables) and Mann–Whitney U-test or Kruskal–Wallis test (continuous variables), as appropriate. Interobserver agreement of semiquantitative PVR assessment was determined by calculation of Cohen’s kappa. Two-sided *p*-values < 0.05 were considered significant. Results for categorical variables are presented as absolute numbers with percentages, and continuous variables are displayed as mean ± standard deviation (SD).

## 3. Results

### 3.1. Clinical Characteristics (Patient-Based Analysis)

Seventy patients were eligible for analysis (95.7% Caucasian descent, 93.3% men, mean age 71 ± 13 years, age range 35–95 years, mean body mass index 28.2 ± 5.5 kg/m^2^). The percentage of patients with diabetes mellitus type 2 was 87.8%. The remaining patients suffered from diabetes mellitus type 1 or pancreatogenic diabetes mellitus. In total, 35 patients were treated with oral antidiabetics (metformin in 26 patients, dipeptidylpeptidase 4-inhibitors in 14 patients, sodium glucose transporter 2-inhibitors in 18 patients, and more than 1 oral antidiabetic drug in 22 patients), and 5 patients were on glucagon-like peptide 1-inhibitors. Forty-three patients were insulin dependent. Seven patients with type 2 diabetes mellitus were treated by diet/lifestyle measures only, and one single patient with diabetes mellitus type 1 did not require insulin treatment after having received a pancreas transplant a few years earlier. Mean HbA1c was 7.7 ± 1.5%, and mean eGFR was 53 ± 29 mL/min/1.73 m^2^. As many as 56.7% of patients suffered from moderate or severe chronic kidney disease (KDIGO categories G3-5, eGFR < 60 mL/min/m^2^), and eight patients were on hemodialysis for end-stage chronic kidney disease. Almost half of the patients (45.6%) had undergone any form of amputation, including toe amputation on the affected leg and/or contralateral minor or major amputation. PAD or peripheral neuropathy was already known in 62.2% and 53.3% of patients, respectively.

### 3.2. Clinical and Hemodynamic Characteristics (Limb-Based Analysis)

In six patients, both legs were analyzed at the same time, and in twelve cases, one leg was analyzed at more than one time point (up to four examinations at different time points), allowing limb-based analysis in a total of ninety cases (forty legs with previous endovascular revascularization). Baseline characteristics are summarized in Table 2.

According to the SINBAD score, 2 cases each were scored with 0, 1, and 6 points, whereas 14, 21, 30, and 19 patients were categorized with 2, 3, 4, and 5 points, respectively. The distribution of different stages of the Wagner–Armstrong classification is depicted in Figure 3. Deep ulcers reaching tendons, joint capsules, or foot bones were present in 52 cases, and 16 cases presented with necrosis. Local infection, as clinically evidenced by redness, swelling, overheating, and purulent wound secretion, was clinically evident in almost two-thirds of cases (n = 59), and accordingly, 65.6% of cases were classified as Wagner–Armstrong categories B or D. Ten patients initially presented with bacteremia/sepsis originating from the diabetic foot wound. Wound swabs were positive in 48 cases, with *Staphylococcus aureus* being by far the most common pathogen, found in 21 cases. Ischemia was clinically suspected in slightly more than half of the cases (n = 47).

Both foot pulses and only one foot pulse were palpable in 18 and 9 legs, respectively, leaving 63 legs (70% of the overall cohort) without palpable foot pulses. ABI measurements were performed in 87 cases. Only 21.1% had ankle brachial index values below 0.9, but more than half of the cases (54.4%) had abnormally elevated ankle brachial index values ≥ 1.3, indicating MAC.

### 3.3. Angiographic Characteristics (Limb-Based Analysis)

Severity of lower extremity arterial obstructions according to the GLASS classification was classified as stage I, II, and III in 24.4%, 31.1%, and 15.1% of cases, respectively. Femoropopliteal and tibial obstructions were observed in 40% and 52.2% of cases, respectively. Combined femoropopliteal and infrapopliteal disease was present in 23.3% of cases. Twenty-six cases had no significantly impaired foot perfusion on angiography (patent femoropopliteal axis and at least one tibial vessel running to the foot without obstructions). Eleven of these cases had stenosis or occlusion of one tibial artery, leaving only fourteen cases with two patent tibial vessels and intact femoropopliteal inflow. Thirteen cases had two occluding tibial vessels with the peroneal artery as the only remaining artery maintaining foot perfusion, seven of whom had a patent peroneal artery with stenosis or occlusion. Overall, the number of diseased arteries below the knee was 0, 1, 2, and 3 in 14, 18, 26, and 32 cases, respectively. The inframalleolar outflow was normal (P0) in 55.6% of cases, moderately impaired (P1) in 34.4% of cases, and severely impaired (P2) in 10% of cases.

### 3.4. Pulse Volume Recordings (Limb-Based Analysis)

#### 3.4.1. Visual Semiquantitative Analysis

PVRs of the forefoot were classified as normal, mildly abnormal, or severely abnormal in 38.9%, 22.2%, and 32.2%, respectively. PVR was non-pulsatile in 6.7% of cases, indicating CLTI. Mean ABI was similar in cases with normal PVR (1.22 ± 0.17) and mildly abnormal PVR (1.22 ± 0.24) but was lower when PVR was severely abnormal (0.94 ± 0.41). The rates of cases with an ABI below 0.5 or with an ABI ≥ 1.3 were 0% and 58.8% for normal PVR, 5% and 85% for mildly abnormal PVR, and 22.2% and 44.4% for severely abnormal PVR. In each two out of the six legs with non-pulsatile PVRs, ABI was 0.2, 0.6, and ≥ 1.3, respectively. Categories of semiquantitative PVR analysis in relation to different grades of PAD severity according to GLASS classification are given in Figure 4. An impaired inframalleolar circulation (inframalleolar GLASS descriptor P1 or P2) was present in 28.6%, 50%, 51.7%, and 83.3% in cases with normal, mildly abnormal, severely abnormal, and non-pulsatile PVRs, respectively.

Among the 84 cases with pulsatile PVRs, an abnormal PVR curve morphology (mildly or severely abnormal) yielded a sensitivity and specificity of 63.3% and 85.7% for detection of severe PAD (GLASS stages 2 and 3). A severely abnormal or non-pulsatile PVR yielded a sensitivity of 34.3% and a specificity of 96.4% for detection of very severe PAD (GLASS 3).

Interobserver agreement of semiquantitative PVR analysis was tested in 36 cases, 15 of whom had repeated assessment after angiography, allowing comparison of 51 pairs of semiquantitative PVR ratings. The ratings were concordant in 44 cases (86%). Discordant findings in seven cases (15.4%) were mainly related to disagreement between normal and mildly abnormal PVRs. Interobserver agreement was substantial (Cohen’s kappa 0.80).

#### 3.4.2. Quantitative Parameters

For quantitative analysis, the six legs with CLTI and non-pulsatile PVRs were also excluded. The mean values (±SD) for UST, USR, and MSA in the 84 analyzed cases were 260 ± 53 ms, 0.32 ± 0.07, and 0.50 ± 0.54 mmHg. The mean values of each of these variables in relation to the GLASS category and stratified according to the inframalleolar descriptor are listed in Table 3 and depicted in Figure 5.

When analyzing the diagnostic accuracy of the quantitative variables for detection of any PAD (GLASS ≥ 1) or for severe PAD (GLASS ≥ 2), ROC analysis revealed the highest area under the curve (AUC) for MSA (AUC 0.89 [95% CI 0.81–0.97] and 0.82 [95% CI 0.73–0.91]). UST (AUC 0.82 [95% CI 0.73–0.91] and 0.71 [95% CI 0.60–0.83]) and USR (AUC 0.81 [95% CI 0.71–0.92] and 0.76 [95% CI 0.65–0.86]) both had worse diagnostic accuracy. With a cut-off value of 0.58 mmHg, MSA had a sensitivity of 91.4% and a specificity of 80.8% for detection of any PAD (GLASS ≥ 1). For detection of severe PAD (GLASS ≥ 2), MSA with a cut-off of 0.27 mmHg had a sensitivity of 72.2% and a specificity of 77.1% (Figure 6).

In a patient-based sensitivity analysis that included only the first examination in patients who were examined at more than one time point (70 patients in total; 65 patients with pulsatile PVRs), the AUC for detection of any PAD (GLASS ≥ 1) or for severe PAD (GLASS ≥ 2) was 0.87 [95% CI 0.77–0.97] and 0.83 [95% CI 0.73–0.92] for MSA, 0.79 [95% CI 0.68–0.90] and 0.69 [95% CI 0.56–0.82] for UST, and 0.80 [95% CI 0.68–0.92] and 0.74 [95% CI 0.61–0.86] for USR.

The mean values of UST and USR did not differ significantly between inframalleolar descriptor categories (UST: P0 257 ± 40 ms, P1 265 ± 59 ms, P2 264 ± 70 ms; USR: P0 0.32 ± 0.07, P1 0.33 ± 0.07, P2 0.35 ± 0.04; *p* = 0.68 and 0.44, respectively). Of note, the mean value of MSA was significantly lower in legs with moderately or severely impaired inframalleolar circulation (P0: 0.63 ± 0.64 mmHg vs. P1/2: 0.32 ± 0.30 mmHg, *p* < 0.01). However, the diagnostic accuracy of MSA for detection of inframalleolar disease was low (cut-off: 0.27 mmHg, sensitivity: 62.9%, and specificity: 69.4%).

#### 3.4.3. Subgroup Analysis: Quantitative Parameters in Legs with ABI Values ≥ 1.3

In 47 cases with an ABI ≥ 1.3 and pulsatile PVRs, the mean values (±SD) for UST, USR, and MSA in these 49 cases were 255 ± 48 ms, 0.32 ± 0.07, and 0.54 ± 0.61 mmHg. Moreover, 16 cases had no significant PAD, and 18, 11, and 2 cases had PAD, GLASS stage I, III, and III, respectively, on angiography.

For detection of any PAD (GLASS ≥ 1) or severe PAD (GLASS ≥ 2) in this subgroup, ROC analysis revealed again the highest AUC for MSA (AUC 0.9 [95% CI 0.78–1.0] and 0.8 [95% CI 0.67–0.93]). UST (AUC 0.75 [95% CI 0.6–0.9] and 0.68 [95% CI 0.49–0.68]) and USR (AUC 0.79 [95% CI 0.65–0.93] and 0.74 [95% CI 0.58–0.9]) performed worse. With a cut-off value of 0.59 mmHg, MSA had a sensitivity of 96.8% and a specificity of 87.5% for detection of any PAD (GLASS ≥ 1). For detection of severe PAD (GLASS ≥ 2), MSA with a cut-off of 0.27 mmHg had a sensitivity of 69.2% and a specificity of 70.6% (Figure 4).

No significant differences were found for mean values of UST and USR in legs with and without impaired inframalleolar outflow. Mean MSA was significantly lower in cases with moderately or severely impaired inframalleolar circulation (P0: 0.71 ± 0.76 vs. P1/2: 0.37 ± 0.34, *p* = 0.02). However, the diagnostic accuracy of MSA for detection of inframalleolar disease was low (cut-off: 0.27 mmHg, sensitivity: 60.9%, and specificity: 69.2%).

#### 3.4.4. Subgroup Analysis: Quantitative Parameters in Limbs Affected by Foot Infection

Fifty-four cases had clinical evidence of foot infection and a pulsatile PVR. The mean values (±SD) for UST, USR, and MSA in these 54 cases were 255 ± 46 ms, 0.32 ± 0.06, and 0.57 ± 0.62 mmHg. Within this subgroup, 19 cases had no significant PAD, whereas 15, 17, and 3 cases had GLASS stage I, III, and III, respectively, on angiography.

For detection of any PAD (GLASS ≥ 1) or for severe PAD (GLASS ≥ 2) in this subgroup, once more MSA had the highest AUC (AUC 0.9 [95% CI 0.8–1.0] and 0.85 [95% CI 0.75–0.95]) and performed better than UST (AUC 0.88 [95% CI 0.78–0.98] and 0.75 [95% CI 0.61–0.88]) and USR (AUC 0.86 [95% CI 0.75–0.96] and 0.75 [95% CI 0.62–0.88]). With a cut-off value of 0.57 mmHg, MSA had a sensitivity of 91.4% and a specificity of 84.2% for detection of any PAD (GLASS ≥ 1). For detection of severe PAD (GLASS ≥ 2), MSA with a cut-off of 0.26 mmHg had a sensitivity of 75.0% and a specificity of 79.4% (Figure 4).

As in the above-mentioned subgroup, no significant differences were evident for mean values of UST and USR in legs with and without impaired inframalleolar outflow. Mean MSA was significantly lower in cases with moderately or severely impaired inframalleolar circulation (P0: 0.69 ± 0.70 vs. P1/2: 0.32 ± 0.31, *p* = 0.01). However, the diagnostic accuracy of MSA for detection of inframalleolar disease was low also in this subgroup (cut-off: 0.27 mmHg, sensitivity: 64.7%, and specificity: 70.3%).

#### 3.4.5. Sensitivity to Change After Revascularization

In forty-four cases, repeated PVRs after endovascular revascularization procedures were analyzed. In cases with improved foot perfusion (n = 35), as documented by final angiography after angioplasty with or without stenting of the femoropopliteal and/or cruropedal arteries, significant improvement of at least one semiquantitative scoring category (e.g., change from mildly abnormal to normal) was seen in only 60%. In eight out of nine legs with failed revascularization attempts, semiquantitative PVR scoring remained unchanged or worsened.

Significant changes were seen for mean MSA after successful revascularization (+0.24 ± 0.26 mmHg), whereas the mean change in MSA was −0.07 ± 1.1 mmHg in nine legs with failed revascularization attempts (*p* < 0.01). An increase in MSA of at least 0.1 mmHg had a sensitivity of 65.7% and a specificity of 88.9% for diagnosis of foot perfusion improvement, as judged by angiography. No significant differences between patients with and without improved foot perfusion after revascularization were found for UST (−34 ± 65 ms vs. −9 ± 56 ms, *p* = 0.47) and USR (−0.02 ± 0.17 vs. −0.01 ± 0.04, *p* = 0.34).

## 4. Discussion

The present study shows promising diagnostic accuracy of quantitative PVRs at the forefoot level for detection of significant PAD in patients suffering from DFS, as verified by invasive angiography. MSA showed better sensitivity and specificity than USR and UST. However, the discriminatory ability of MSA between severe and very severe PAD was only modest, as was its predictive value for detection of an impaired outflow below the ankle. Subgroup analyses showed consistent findings for patients with elevated ankle brachial index due to MAC and for patients suffering from infected DFS (Armstrong categories B and D). Semiquantitative PVR assessment showed substantial interobserver agreement; however, it also showed limited sensitivity for detection of severe and very severe PADs.

PAD is common in patients suffering from DFS (up to 50%). In diabetic patients, PAD frequently has a distal distribution pattern with obstructive lesions of the crural and/or pedal arteries [15]. While primarily ischemic wounds represent only a minority of DFS cases, sufficient foot perfusion is considered to be crucial for wound healing in all types of DFS [5]. Angiography as an invasive procedure with arterial punctures carries a non-negligible complication risk. Therefore, noninvasive assessment of foot perfusion in DFS is of utmost clinical importance but remains challenging. This is mainly related to the high prevalence of MAC of the below-the-knee and below-the-ankle arteries in diabetic patients, making systolic ankle pressure measurements and ABI unreliable for the diagnosis of PAD. Besides being a strong predictor of cardiovascular mortality, MAC is associated with distal distribution of arteriosclerosis and the development of CLTI [16]. Moreover, MAC is associated with peripheral neuropathy in diabetic patients, probably related to sympathetic denervation of the smooth muscle cells of the tunica media [17,18].

Current guidelines recommend several alternative noninvasive tests for detection of PAD in this clinical scenario, clinically characterized by absent foot pulses [5]. Among these, systolic toe pressures with calculation of the toe–brachial index (TBI) and analysis of pedal Doppler waveforms are the best evaluated diagnostic strategies [6]. A toe brachial index > 0.7 has moderate ability to rule out significant PAD, whereas an absolute toe pressure > 30 mmHg has been suggested to substantially increase the chance of wound healing of DFS [5,6]. Normahani et al. compared several noninvasive diagnostic strategies and found pedal Doppler waveform analysis to have the highest sensitivity for detection of PAD [19]. In a prospective follow-up of the aforementioned study, the highest negative predictive values for wound healing also were seen with monophasic pedal Doppler waveforms [20].

The first pulse volume recorder was developed at the Massachusetts General Hospital in 1972 by Darling et al. [21]. Back in 1986, PVRs were included in the Rutherford classification as a diagnostic criterion for CLTI [8]. Currently, PVRs are frequently used in vascular medicine to assess arterial blood flow in the lower extremities, particularly in the scenario of incompressible ankle arteries due to MAC. However, scientific evidence regarding the diagnostic accuracy of PVRs for detection and severity grading of PAD is surprisingly poor, with most of the few available studies performed and published several decades ago [22,23,24]. Very few studies compared PVRs with other noninvasive measures for detection of PAD in patients with DFS. Randhawa et al. compared TBIs and PVRs for detection of tibial artery disease, as assessed by angiography, in patients with suspected CLTI and noncompressible tibial arteries (87.6% diabetic patients) [25]. The authors stated that the sensitivity for detecting significant tibial artery disease was much higher for the TBI (89.7%) compared to PVR (43.6%). However, in that study PVR recordings were obtained at the ankle, and a substantial proportion of patients had additional inframalleolar disease with lost pedal arch patency. Therefore, in our opinion, the sensitivity of PVR would have been much higher when cuffs would have been placed at the metatarsal level [26]. In a study on 303 patients with type 2 diabetes mellitus, PVR (sensitivity 83.8%, specificity 93.3%) had higher diagnostic accuracy for detection of PAD, as defined by color duplex ultrasonography of the infrainguinal arteries, compared to the ABI (cut-off < 0.9, sensitivity 72.7%, specificity 95.8%), but lower diagnostic accuracy compared to the TBI (cut-off < 0.38, sensitivity 100%, specificity 95.8%) [27].

In the aforementioned studies and in routine clinical practice, PVR assessment relies on semiquantitative analysis of PVR curve morphology, as outlined by the guidelines for noninvasive vascular laboratory testing [14]. Just recently, some authors performed, for the first time, quantitative analysis of PVRs with respect to the UST and USR [28,29]. These studies revealed a negative correlation between PVR-derived USR and ankle/toe brachial index as well as a good responsiveness to revascularization effects and substantial inter-rater reliability. According to our results, we strongly propose quantitative PVR assessment of forefoot perfusion (MSA preferable over UST and USR) as a useful diagnostic tool for the detection of PAD in patients with DFS, including those with falsely elevated ABI and those suffering from foot infection.

Strengths of our study include the well-defined study population covering the whole spectrum of the DFS, the considerable sample size, and the comparison with the gold standard of invasive angiography (with structured assessment according to the GLASS classification). Several limitations must be acknowledged. As we did not routinely perform toe pressure measurements, we were not able to calculate the TBI and the WIFi scores, which are suggested by recent guidelines [4]. We further did not routinely assess pedal Doppler waveforms, prohibiting us from comparative analysis of different noninvasive diagnostic methods. A structured diagnostic workup of peripheral neuropathy and of MAC scoring in plain radiographs was also not performed in our retrospective study. Therefore, prospective studies are warranted to address the above-mentioned limitations. These should include comparative analysis of the different noninvasive methods currently available against the gold standard of DSA as well as longitudinal follow-up of the patients in order to assess their predictive value of forefoot PVRs with quantitative assessment for wound healing in DFS. Furthermore, it remains to be clarified whether the combination of diagnostic strategies (e.g., PVR plus TBI) can offer a higher diagnostic and prognostic yield compared to the single methods.

## 5. Conclusions

PVRs at the forefoot level show promising diagnostic accuracy for detection of PAD in patients suffering from DFS, irrespective of MAC. Semiquantitative assessment (sensitivity and specificity of 63.3% and 85.7% for detection of severe PAD) was less sensitive but more specific than quantitative measurements, and out of all quantitative parameters assessed, MSA (with a cut-off value of 0.27 mmHg sensitivity and specificity of 72.2% and 77.1% for detection of severe PAD) outperformed UST and USR. Our results were consistent in the subgroups of patients with mediasclerotic ankle pressures and with foot infection. However, PVRs at the forefoot level have limited ability for diagnosis of inframalleolar disease.

## Figures and Tables

**Figure 1 biomedicines-13-01281-f001:**
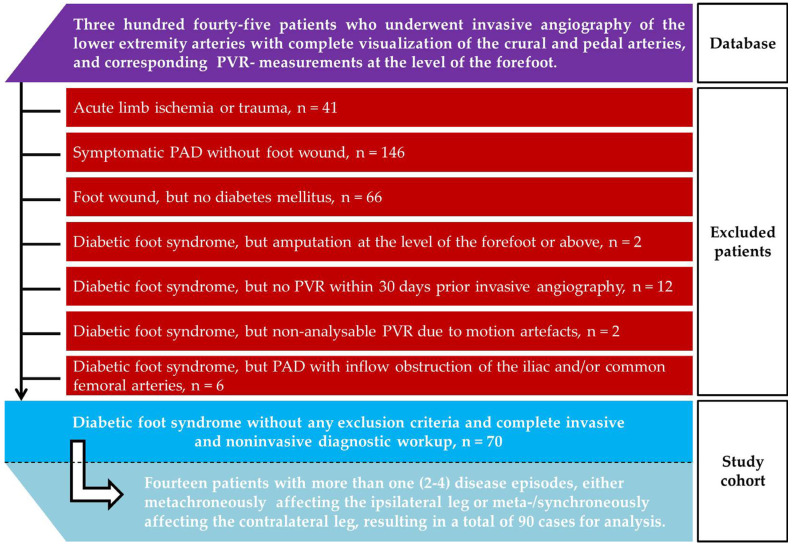
Flowchart of patient identification for the present study.

**Figure 2 biomedicines-13-01281-f002:**
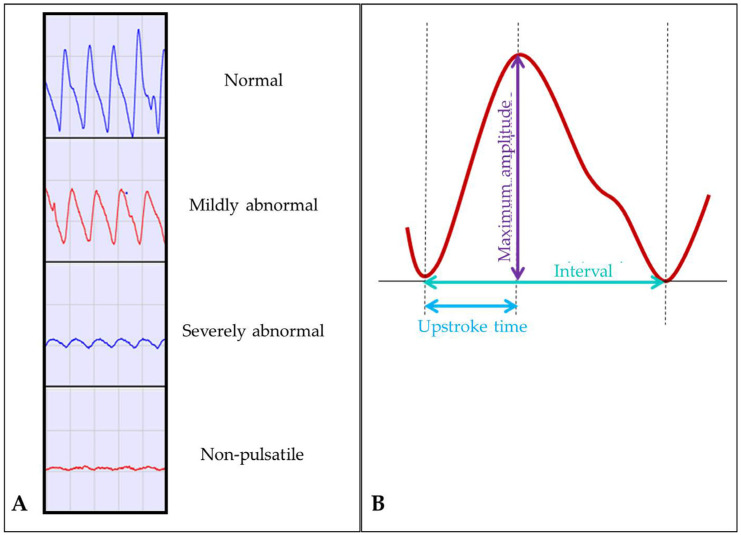
Categories of visual semiquantitative grading of PVRs with examples (**A**) and illustration of quantitative measures of PVRs (**B**).

**Figure 3 biomedicines-13-01281-f003:**
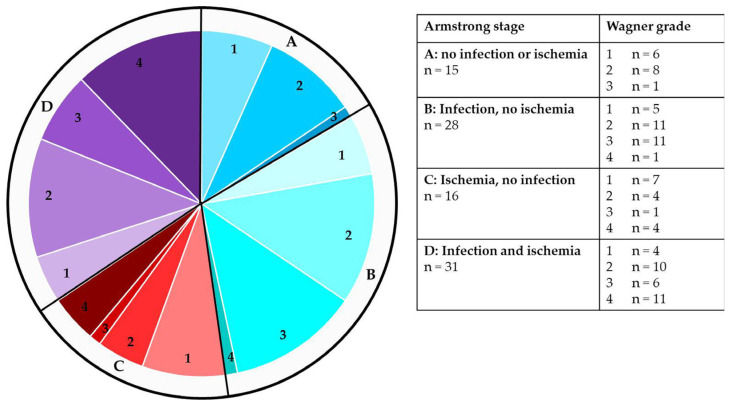
Distribution of different categories of the Wagner–Armstrong classification in 90 extremities of 70 patients with diabetic foot syndrome.

**Figure 4 biomedicines-13-01281-f004:**
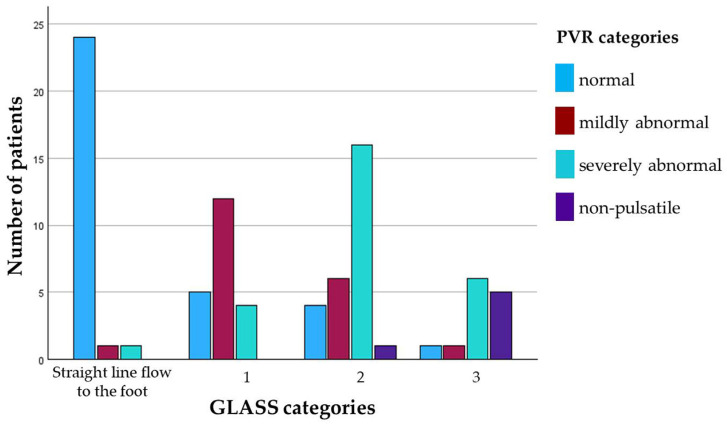
The relationship between semiquantitatively assessed PVR morphology and severity of PAD according to the GLASS classification.

**Figure 5 biomedicines-13-01281-f005:**
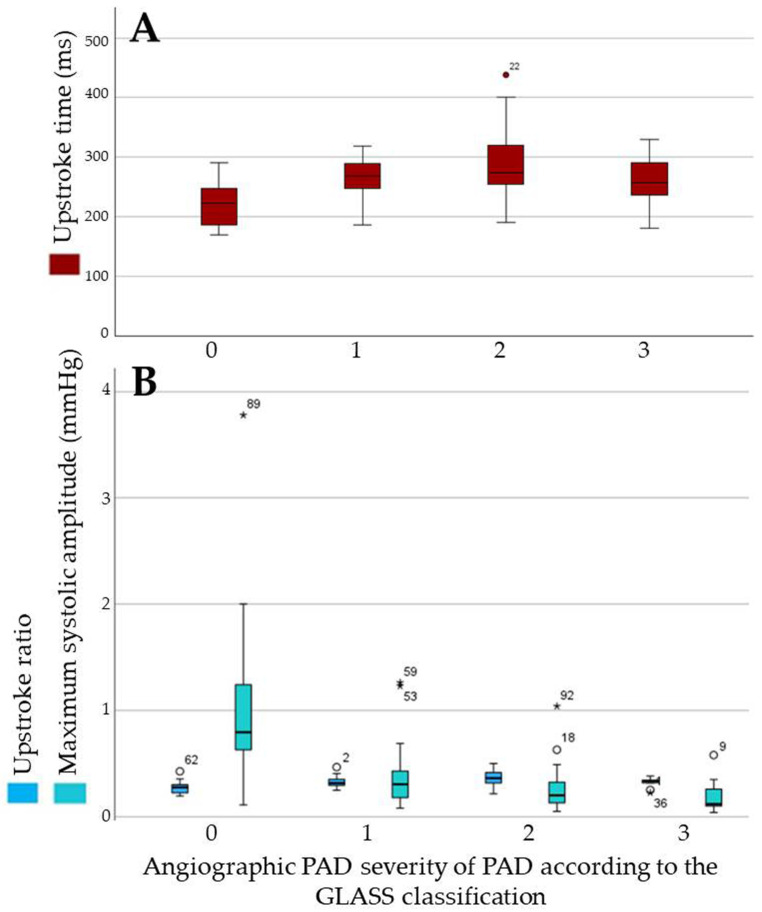
Median, maximum, and minimum values together with interquartile ranges of upstroke time (**A**), upstroke ratio, and maximum systolic amplitude (**B**) in different categories of the GLASS classification.

**Figure 6 biomedicines-13-01281-f006:**
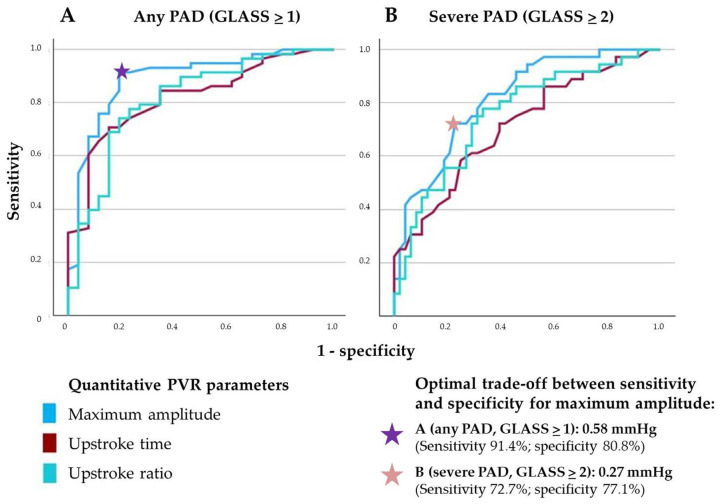
Diagnostic accuracy, as depicted by ROC analysis of quantitative parameters of PVRs for detection of any PAD (**A**) and severe PAD (**B**) in the overall cohort.

**Table 1 biomedicines-13-01281-t001:** Inclusion and exclusion criteria.

Inclusion Criteria	Exclusion Criteria
Patients with diabetes mellitus suffering from nontraumatic foot wounds	Patients with diabetic foot syndrome, Wagner stage 5 (necrosis of the complete foot)
Invasive angiography with complete depiction of the femoral, popliteal, below-the-knee, and below-the-ankle arteries	Patients with peripheral arterial disease without foot wounds, Rutherford categories I–IV
Pulse volume recordings of the forefoot within 4 weeks prior to invasive angiography	Patients with peripheral arterial disease affecting the iliac artery inflow
	Patients with nonatherosclerotic peripheral arterial disease
	Patients with acute limb ischemia
	Patients with nontraumatic foot wounds not suffering from diabetes mellitus
	Patients suffering from traumatic foot wounds
	Patients with prior amputation proximal to the toe level
	Patients with non-analyzable PVRs due to motion artifacts

**Table 2 biomedicines-13-01281-t002:** Baseline characteristics of the study cohort (limb-based analysis).

Variable	Limb-Based Analysis n = 90
Known peripheral arterial disease, n (%)	56 (62.2)
Prior lower extremity artery revascularization, n (%)	40 (44.4)
Known peripheral neuropathy, n (%)	49 (54.4)
Righ/left leg affected by diabetic foot ulcer, n (%)	44 (48.9)/46 (51.1)
Diabetic foot ulcer duration, weeks (mean ± SD)	18.9 ± 39.3
Foot pulses absent in the affected leg, n (%)	63 (70)
Clinical suspicion of ischemia, n (%)	47 (52.2)
ABI < 0.9/≥1.3, n (%) *	19 (21.1)/49 (54.4)
Clinical signs and symptoms of infection, n (%)	59 (65.6%)
Wound swab positive, n (%)	48 (53.3)
Radiographic suspicion of osteomyelitis, n (%)	30 (66.7)

* Systolic ankle pressure measurements were performed in 87 of 90 legs.

**Table 3 biomedicines-13-01281-t003:** Different degrees of severity of PAD and the inframalleolar outflow according to the GLASS classification, with corresponding quantitative values of PVR measurements at the level of the forefoot.

IMD	Straight-Line Inflow to the Foot	GLASS I	GLASS II	GLASS III
**P0**	n = 18	n = 13	n = 16 *	n = 3
	UST 221 ± 30 ms	UST 271 ± 38 ms	UST 291 ± 51 ms	UST 236 ± 22 ms
	USR 0.28 ± 0.05	USR 0.32 ± 0.03	USR 0.37 ± 0.08	USR 0.29 ± 0.06
	MSA 1.10 ± 0.78 mmHg	MSA 0.47 ± 0.38 mmHg	MSA 0.31 ± 0.26 mmHg	MSA 0.16 ± 0.16 mmHg
**P1**	n = 7	n = 7	n = 9	n = 11 ^#^
	UST 215 ± 41 ms	UST 264 ± 38 ms	UST 296 ± 68 ms	UST 276 ± 53 ms
	USR 0.25 ± 0.03	USR 0.33 ± 0.08	USR 0.37 ± 0.06	USR 0.33 ± 0.04
	MSA 0.66 ± 0.48 mmHg	MSA 0.30 ± 0.14 mmHg	MSA 0.29 ± 0.05 mmHg	MSA 0.21 ± 0.19 mmHg
**P2**	n = 1	n = 2	n = 3	n = 0
	UST 291 ms	UST 231 ± 46 ms	UST 277 ± 97 ms	
	USR 0.43	USR 0.34 ± 0.11	USR 0.33 ± 0.03	
	MSA 0.65 mmHg	MSA 0.17 ± 0.01 mmHg	MSA 0.29 ± 0.05 mmHg	

* One patient with non-pulsatile PVRs, indicating CLTI. ^#^ Five patients with non-pulsatile PVRs, indicating CLTI. IMD, inframalleolar descriptor; UST, upstroke time; USR, upstroke ratio; MSA, maximum systolic amplitude; CLTI, critical limb-threatening ischemia.

## Data Availability

The data presented in this study are available upon request from the corresponding author.

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
