# Peer review of "Segmental Pulse Volume Recordings at the Forefoot Level as a Valuable Diagnostic Tool for Detection of Peripheral Arterial Disease in the Diabetic Foot Syndrome"

_biomedicines, 2025, doi:10.3390/biomedicines13061281_

Round 1

Reviewer 1 Report (Previous Reviewer 1)

Comments and Suggestions for Authors

The authors made significant revisions based on the reviewers' comments; the manuscript is now acceptable for publication.

Author Response

Thank you very much!

Reviewer 2 Report (Previous Reviewer 2)

Comments and Suggestions for Authors

The authors made several revisions asked by the reviewers.

Here are my comments :

  • Table 1 and figure 1 present some featured datas. Figure 1 could be simplified for more clarity.
  • Have the authors included all types of diabetes ? How the specific types were diagnosed ( genetic sampling ? )
  • Improve figure 3 and figure 6
  • results are too long and could be more synthetized with tables and figures.
  • Conclusion must summarize main results.

Author Response

2.1. Table 1 and figure 1 present some featured datas. Figure 1 could be simplified for more clarity.

The detailed workflow of patient selection/exclusion was explicitly requested by some of the reviewers in review round. Therefore, we see no chance for simplification of the Figure 1 which we added to the manuscript after the first review round.

2.2. Have the authors included all types of diabetes ? How the specific types were diagnosed (genetic sampling ?

Indeed, we included all types of diabetes mellitus and classification was done based on the information in the medical records according to current recommendations (ADA), as stated in the results section. Genetic testing was not done, as not routinely required for the clinical classification and not feasible in this retrospective study.

2.3. Improve figure 3 and figure 6

This was due to formatting problems of the Journal`s template. We reformatted the revised manuscript.

2.4. results are too long and could be more synthetized with tables and figures.

During review round 1, we had to address exceptionally many reviewer comments, with several requests for additional analyses and information in the results section which we provided accordingly. At its current status, the manuscript contains 6 figures and 3 tables, and in our opinion additional figures/tables would lead to an overload of the paper. We carefully revised the results section in order to reduce the word count, but within the context of the above-mentioned previous reviewer requests we are not able to remove complete text passages from the results section.

2.5. Conclusion must summarize main results.

We revised the conclusion according to this comment.

Reviewer 3 Report (Previous Reviewer 3)

Comments and Suggestions for Authors

Authors revised the manuscript, but this article has not fully answered some of the questions due to insufficient description.
First, authors suggest “Interobserver agreement was substantial (Cohen`s kappa 0.8).” (L30), but it is difficult to understand what authors explain as a sentence in abstract. “Interobserver agreement was substantial (Cohen`s kappa 0.8).” (L30) may be ”Interobserver agreement of semiquantitative PVR rating was substantial (Cohen`s kappa 0.8) among 51 cases”.
Second, authors suggest “We identified all patients who underwent digital subtraction angiography of the lower extremity arteries between 01/2020 and 11/2024.” (L74), but it is difficult to understand how to gather the participants of this study and what the characteristics of participants (e.g., nationality and range of age). Authors should explain how to gather them as well as their characteristics in method section.
Third, authors suggest “A normal forefoot PVR was defined by a sharp upstroke, a scooped or flat interval between peaks and a dicrotic notch at the diastolic descending part of the curve. PVR was classified as mildly abnormal when the dicrotic notch and the flat interval between curves were lost, while the upstroke remained steep. PVR was classified as severely abnormal when in addition to the above mentioned changes the amplitude was flattened and the upstroke got prolonged (equal upstroke and downslope period).” (L139), but they do not explain them using quantitative definition. For example, they did not define “sharp”, “scooped”, “flat”, “dicrotic”, “steep”, “flattened”, “prolonged”, and “equal”, quantitatively. Without quantity indicators, it is difficult to identify them, scientifically. Authors should add quantitative definition in method section.
Fourth, authors suggest “Wound swabs were positive in 48 cases, with staphylococcus aureus being by far the most common pathogen, found in 21 cases.” (L225), but they do not explain method to identify “staphylococcus aureus” from would swabs (e.g., name of ELIZA kit). It is difficult to understand what authors did, without explanation. Authors should add the explanation in method section.
Finally, authors used Figure 3, but it is difficult to read the explanation. Authors should revise Figure 3.

Author Response

Reviewer 3

3.1. First, authors suggest “Interobserver agreement was substantial (Cohen`s kappa 0.8).” (L30), but it is difficult to understand what authors explain as a sentence in abstract. “Interobserver agreement was substantial (Cohen`s kappa 0.8).” (L30) may be ”Interobserver agreement of semiquantitative PVR rating was substantial (Cohen`s kappa 0.8) among 51 cases”.

We fully agree and changed the abstract according to this useful advice.

3.2. Second, authors suggest “We identified all patients who underwent digital subtraction angiography of the lower extremity arteries between 01/2020 and 11/2024.” (L74), but it is difficult to understand how to gather the participants of this study and what the characteristics of participants (e.g., nationality and range of age). Authors should explain how to gather them as well as their characteristics in method section.

In this case, we disagree. In the revised manuscript, we explain the process of patient selection in detail and also provide detailed clinical characteristics of the included patients. The demographic and ethnic characteristics of excluded patients do not offer a scientific benefit within the context of this paper. Therefore, as long as not otherwise requested by the Editor, we decided not to change the manuscript with respect to this comment.

3.3. Third, authors suggest “A normal forefoot PVR was defined by a sharp upstroke, a scooped or flat interval between peaks and a dicrotic notch at the diastolic descending part of the curve. PVR was classified as mildly abnormal when the dicrotic notch and the flat interval between curves were lost, while the upstroke remained steep. PVR was classified as severely abnormal when in addition to the above mentioned changes the amplitude was flattened and the upstroke got prolonged (equal upstroke and downslope period).” (L139), but they do not explain them using quantitative definition. For example, they did not define “sharp”, “scooped”, “flat”, “dicrotic”, “steep”, “flattened”, “prolonged”, and “equal”, quantitatively. Without quantity indicators, it is difficult to identify them, scientifically. Authors should add quantitative definition in method section.

As stated already in the first review round, the mentioned classification of PVR is performed semiquantitatively, as frequently done in all fields of Clinical Medicine and as suggested by specific recommendations also cited in the manuscript [Gerard-Herman et al. Vasc Med. 2006 Nov;11(3):183-200. doi: 10.1177/1358863x06070516]. In addition to this semiquantitative analysis we performed detailed quantitative analysis and meticulously described the underlying methodology.

3.4. Fourth, authors suggest “Wound swabs were positive in 48 cases, with staphylococcus aureus being by far the most common pathogen, found in 21 cases.” (L225), but they do not explain method to identify “staphylococcus aureus” from would swabs (e.g., name of ELIZA kit).

Bacterial infection is not the scientific focus of this paper. However, we included additional information in the methods section.

3.5. It is difficult to understand what authors did, without explanation. Authors should add the explanation in method section.

It is not clear which part of our paper this sentence refers to. In case of specific recommendations for improvement of the manuscript, we are willing to perform additional changes.

3.6. Finally, authors used Figure 3, but it is difficult to read the explanation. Authors should revise Figure 3.

This was due to formatting problems of the Journal`s template. We reformatted the revised manuscript.

Reviewer 4 Report (Previous Reviewer 4)

Comments and Suggestions for Authors

In my view, the revised version is ready for publication. 

Author Response

Thank you very much!

Round 2

Reviewer 3 Report (Previous Reviewer 3)

Comments and Suggestions for Authors

Authors revised the manuscript, but this article has not fully answered some of the questions due to insufficient description.
First, as mentioned in the previous reviews, authors suggest “We identified all patients who underwent digital subtraction angiography of the lower extremity arteries with complete visualization of the crural and pedal arteries between 01/2020 and 11/2024.” (L70), but it is difficult to understand how to gather the participants of this study and what the characteristics of participants (e.g., nationality and range of age). Authors suggest “In this case, we disagree. In the revised manuscript, we explain the process of patient selection in detail and also provide detailed clinical characteristics of the included patients. The demographic and ethnic characteristics of excluded patients do not offer a scientific benefit within the context of this paper. Therefore, as long as not otherwise requested by the Editor, we decided not to change the manuscript with respect to this comment.”, but without the detailed explanations, it is difficult to rule out selection bias, and proving reproducibility is also difficult. Moreover, there may be racial differences in the results, but without information such as nationality, readers cannot take this into account. Authors should explain how to gather them as well as their characteristics in method section.
Second, as mentioned in the previous reviews, authors suggest “PVR were assessed semiquantitatively with regard to curve morphology (side and level differences of amplitude, systolic upstroke and diastolic dicrotic notch) (Figure 2,A). A normal forefoot PVR was defined by a sharp upstroke, a scooped or flat interval between peaks and a dicrotic notch at the diastolic descending part of the curve. PVR was classified as mildly abnormal when the dicrotic notch and the flat interval between curves were lost, while the upstroke remained steep. PVR was classified as severely abnormal when in addition to the above mentioned changes the amplitude was flattened and the upstroke got prolonged (equal upstroke and downslope period).” (L112), but they do not explain them using quantitative definition. For example, they did not define “sharp”, “scooped”, “flat”, “dicrotic”, “steep”, “flattened”, “prolonged”, and “equal”, quantitatively. Without quantity indicators, it is difficult to identify them, scientifically. Authors suggest “As stated already in the first review round, the mentioned classification of PVR is performed semiquantitatively, as frequently done in all fields of Clinical Medicine and as suggested by specific recommendations also cited in the manuscript [Gerard-Herman et al. Vasc Med. 2006 Nov;11(3):183-200. doi: 10.1177/1358863x06070516]. In addition to this semiquantitative analysis we performed detailed quantitative analysis and meticulously described the underlying methodology.”, but they did not explain the shapes, quantitatively, and they have also failed to show how to ensure the reproducibility of measurements. Authors should add quantitative definition in method section.
Finally, as mentioned in the previous reviews, authors suggest “Wound swabs were positive in 48 cases, with staphylococcus aureus being by far the most common pathogen, found in 21 cases.” (L179), but they do not explain method to identify “staphylococcus aureus” from would swabs (e.g., name of ELIZA kit). Authors suggest “Bacterial infection is not the scientific focus of this paper. However, we included additional information in the methods section.” and added the sentence “The bacterial pathogens isolated from the individual wounds were recorded if wound swabs had been taken.” (L94), but they have focused on “infection” as well as ischemia regarding Wagner-Armstrong classification in Figure 3. It is difficult to understand what authors did, without explanation. Authors should add the explanation in method section.

Author Response

Reviewer 3

First, as mentioned in the previous reviews, authors suggest “We identified all patients who underwent digital subtraction angiography of the lower extremity arteries with complete visualization of the crural and pedal arteries between 01/2020 and 11/2024.” (L70), but it is difficult to understand how to gather the participants of this study and what the characteristics of participants (e.g., nationality and range of age).

We added the range of age of included patients. Due to the retrospective nature of the study we were not able to clarify the nationality of all patients without doubt, but we provide information on ethnicity in the revised manuscript.

Authors suggest “In this case, we disagree. In the revised manuscript, we explain the process of patient selection in detail and also provide detailed clinical characteristics of the included patients. The demographic and ethnic characteristics of excluded patients do not offer a scientific benefit within the context of this paper. Therefore, as long as not otherwise requested by the Editor, we decided not to change the manuscript with respect to this comment.”, but without the detailed explanations, it is difficult to rule out selection bias, and proving reproducibility is also difficult. Moreover, there may be racial differences in the results, but without information such as nationality, readers cannot take this into account. Authors should explain how to gather them as well as their characteristics in method section.

As stated above, we now provide information on ethnicity of included patients.

Second, as mentioned in the previous reviews, authors suggest “PVR were assessed semiquantitatively with regard to curve morphology (side and level differences of amplitude, systolic upstroke and diastolic dicrotic notch) (Figure 2,A). A normal forefoot PVR was defined by a sharp upstroke, a scooped or flat interval between peaks and a dicrotic notch at the diastolic descending part of the curve. PVR was classified as mildly abnormal when the dicrotic notch and the flat interval between curves were lost, while the upstroke remained steep. PVR was classified as severely abnormal when in addition to the above mentioned changes the amplitude was flattened and the upstroke got prolonged (equal upstroke and downslope period).” (L112), but they do not explain them using quantitative definition. For example, they did not define “sharp”, “scooped”, “flat”, “dicrotic”, “steep”, “flattened”, “prolonged”, and “equal”, quantitatively. Without quantity indicators, it is difficult to identify them, scientifically.

As stated for several times now: this is a semiquantitative method which we performed in accordance to specific recommendations broadly accepted in the Vascular community [Gerard-Herman et al. Vasc Med. 2006 Nov;11(3):183-200. doi: 10.1177/1358863x06070516]. It is simply not possible to provide a quantitative criterion for a semiquantitative method.

Finally, as mentioned in the previous reviews, authors suggest “Wound swabs were positive in 48 cases, with staphylococcus aureus being by far the most common pathogen, found in 21 cases.” (L179), but they do not explain method to identify “staphylococcus aureus” from would swabs (e.g., name of ELIZA kit). Authors suggest “Bacterial infection is not the scientific focus of this paper. However, we included additional information in the methods section.” and added the sentence “The bacterial pathogens isolated from the individual wounds were recorded if wound swabs had been taken.” (L94), but they have focused on “infection” as well as ischemia regarding Wagner-Armstrong classification in Figure 3. It is difficult to understand what authors did, without explanation. Authors should add the explanation in method section.

According to this comment, we added the principles of characterization of bacterial isolates used in our clinical routine.

This manuscript is a resubmission of an earlier submission. The following is a list of the peer review reports and author responses from that submission.

Round 1

Reviewer 1 Report

Comments and Suggestions for Authors

The article titled “Segmental pulse volume recordings at the forefoot level as a useful diagnostic tool for detection of peripheral arterial disease in the diabetic foot syndrome” has been evaluated. 

In this study, the authors focused on patients diagnosed with Diabetic Foot Syndrome (DFS) who underwent invasive angiography between January 2020 and November 2024. They included only those patients who had a corresponding pulse volume recording (PVR) performed within 30 days prior to their angiographic procedure. To classify the severity of DFS, the Wagner-Armstrong classification system was employed. The authors meticulously recorded clinical characteristics and hemodynamic parameters, which included systolic ankle pressures and the ankle-brachial index (ABI), to assess blood flow in the lower extremities.

The analysis of PVR was performed semiquantitatively by investigators who were blinded to the clinical details of the participants. Additionally, a quantitative assessment was conducted, determining key metrics such as upstroke time (UST), upstroke ratio (USR), and maximum systolic amplitude (MSA). The severity of peripheral arterial disease (PAD) was classified according to the GLASS classification system.

For the statistical evaluation, the researchers employed various methods including univariate significance tests, 2 x 2 contingency tables, receiver operating characteristic (ROC) analysis, and an assessment of interobserver agreement. A total of 90 extremities from 70 patients were analyzed in this study, of which 47 presented with an ABI greater than 1.3. Notably, critical limb-threatening ischemia, characterized by non-pulsatile PVR, was observed in 6.7% of the evaluated cases. 

The analysis revealed that an abnormal PVR curve morphology, categorized as mildly or severely abnormal, demonstrated a sensitivity of 86.1% and a specificity of 62.5% for the detection of severe PAD, corresponding to GLASS stages 2 and 3. The interobserver agreement in interpreting these results was substantial, with a Cohen's kappa value of 0.8, indicating high consistency among different observers.

When evaluating diagnostic accuracy, maximum systolic amplitude (MSA) emerged as the most effective metric. The area under the curve (AUC) for MSA was found to be 0.89 for detecting any PAD (GLASS > 1) and 0.83 for identifying severe PAD (GLASS > 2). Specifically, with a cutoff value of 0.58 mmHg, MSA exhibited a sensitivity of 91.4% and a specificity of 80.8% for detecting any PAD. Moreover, with a lower cutoff of 0.27 mmHg, MSA provided a sensitivity of 72.2% and a specificity of 77.1% for severe PAD, as well as a sensitivity of 62.9% and specificity of 69.4% for identifying inframalleolar disease. These results remained consistent across various subgroup analyses.

In conclusion, the authors assert that PVR, particularly when analyzed through quantitative features, presents a promising diagnostic tool for detecting PAD within the context of DFS. MSA demonstrated superior performance compared to UST and USR, although it exhibited limitations in detecting impaired inframalleolar outflow.

However, before the manuscript can be deemed suitable for publication, substantial revisions are required in several key areas:

1. This section should be expanded to include more statistical data and a thorough discussion on various diagnostic methods available for detecting peripheral arterial disease in diabetic foot syndrome.

2. The authors should incorporate a flow diagram detailing all experimental procedures, including specific inclusion and exclusion criteria for participants in the study.

3. While univariate group comparisons utilized the χ2-test for categorical variables and the Mann-Whitney U-test for continuous variables, the authors should explore additional statistical methods that could enhance the robustness of their data analysis.

4. The conclusion should be elaborated upon, providing a more detailed discussion of the results, their implications, and how they relate to existing literature.

5. The reference list currently lacks depth; the authors should update and expand this section with more relevant and recent references to support their findings and enhance the credibility of their work.

Author Response

1.1.  This section should be expanded to include more statistical data and a thorough discussion on various diagnostic methods available for detecting peripheral arterial disease in diabetic foot syndrome.

To properly address this comment, we broadened the discussion substantially. Furthermore, we added a sensitivity analysis including only the first examination in patients with more than one diagnostic instances (as outlined in detail in 1.3.)

1.2. The authors should incorporate a flow diagram detailing all experimental procedures, including specific inclusion and exclusion criteria for participants in the study.

Done.

1.3. While univariate group comparisons utilized the χ2-test for categorical variables and the Mann-Whitney U-test for continuous variables, the authors should explore additional statistical methods that could enhance the robustness of their data analysis.

With regard to this comment, we sought statistical advice (Dr. Crispin, Institute for Biometrics and Medical Statistics, Ludwig-Maximilians-University, Munich, Germany). He attested us that the statistical methods we applied are sound and appropriate to answer our study questions. Instead of performing additional statistics, he advised us to perform a sensitivity analysis including only one examination per patient to enhance the robustness of our analysis. For further analysis with comparison of more than 2 subgroups, we additionally applied the Kruskal-Wallis-Test.

1.4. The conclusion should be elaborated upon, providing a more detailed discussion of the results, their implications, and how they relate to existing literature.

We agree and revised this text segment, as suggested.

1.5. The reference list currently lacks depth; the authors should update and expand this section with more relevant and recent references to support their findings and enhance the credibility of their work.

We agree and discussed the available evidence in more detail.

Reviewer 2 Report

Comments and Suggestions for Authors

The authors analysed the segmental pulse volume recordings at the forefoot level as a useful diagnostic tool for detection of peripheral arterial disease in the diabetic foot syndrome.

This is an original article , the format is correct and the english spelling is clear.

- Introduction is clear : authors could add more epidemiological data especially on diabetes.

- PVR full spelling is missing

- Add in the methodology if a sampling was calculated. And if the patients were exhaustively choosed.

- classically the IWGDF classification is used. Explain which classifications are used in order to assess the diabetic foot.

- What types of diabetes were choosed, and how the diagnosis was made.

- the most important data to assess is the duration of diabetes. the authors must explain the relation between duration and severity.

- explain the pancreas transplant : because normally this patient is cured and excluded.

- could the authors add more data on treatment. SGLT2-inh are known to increase the risk of arteriopathy in lower limb and could biase the interpretation.

- Tables and figures are clear. However increase the size of Figure 5. If you could add the line to show the best cut-off

- References could be enhanced knowing that there are other recent articles on diabetic foot lesions.

- Add strengths and limitations.

- Add at the end a practical aspects on how to address your results in reality.

Author Response

2.1. Introduction is clear : authors could add more epidemiological data especially on diabetes.

We added additional epidemiological data.

2.2. PVR full spelling is missing

Corrected.

2.3. Add in the methodology if a sampling was calculated. And if the patients were exhaustively choosed.

As this is a retrospective analysis sample size estimation was not performed. We added this information in the revised manuscript, which also contains a flow chart depicting the patient selection process (as suggested by Reviewer 1, see our response to comment 1.2.)

2.4. classically the IWGDF classification is used. Explain which classifications are used in order to assess the diabetic foot.

To address this comment, we additionally categorized our patients according to the SINBAD-classification, as suggested by the IWGDF.

2.5. What types of diabetes were choosed, and how the diagnosis was made.

We give this information in the methods section.

2.6. the most important data to assess is the duration of diabetes. the authors must explain the relation between duration and severity.

As this is a retrospective study, exact information on diabetes duration was lacking in several patients. Since the diabetes duration is not primarily relevant for the diagnostic accuracy of a diagnostic method, we believe that this information is not necessary to adequately answer our study questions.

2.7. explain the pancreas transplant : because normally this patient is cured and excluded.

Although this patient did not require further insulin treatment after transplantation, he suffered from severe late complications of type 1 diabetes mellitus, including diabetic nephropathy (eGFR 50 ml/min), neuropathy and peripheral arterial disease (toe amputation on the contralateral leg). As our study focuses on the diagnosis of a diabetes complication and not on diabetes mellitus itself, we would like to keep the patient in our analysis.

2.8. could the authors add more data on treatment. SGLT2-inh are known to increase the risk of arteriopathy in lower limb and could biase the interpretation.

Thank you for this important comment. We retrieved additional data on SGLT2i and other oral antidiabetics as well as GLP1a. We added this information in the manuscript.

2.9. Tables and figures are clear. However increase the size of Figure 5. If you could add the line to show the best cut-off

Done.

2.10. References could be enhanced knowing that there are other recent articles on diabetic foot lesions.

We agree and discussed the available evidence on diabetic foot lesions in more detail.

2.11. Add strengths and limitations.

Strenghts and limitations are already listed in our manuscript.

2.12. Add at the end a practical aspects on how to address your results in reality.

We agree and give practical recommendations based on our results.

Reviewer 3 Report

Comments and Suggestions for Authors

Authors used a cross-sectional study to evaluate diagnosis by pulse volume recordings on identification of peripheral arterial disease among patients with diabetic foot syndrome. However, this article has not fully answered some of the questions due to insufficient description and inadequate statistical analysis.

First, authors suggest “There were significant differences in the intervention duration, the dose of the intervention, and the regions of the intervention compared with the control group.” (L205), but they do not describe statistical results. Without statistical results, it is difficult to understand what they explain. Authors should revise result session, carefully.

Second, authors suggest “Interobserver agreement was substantial” (L30), “for detection of inframalleolar disease” (L35), and “An impaired inframalleolar circulation (inframalleolar GLASS descriptor P1 or P2) was present in 28.6%, 50%, 51.7% and 83.3%” (L242), “Interobserver agreement of PVR was tested in 36 cases” (L246), but it is difficult to understand what authors explain. Authors should revise the manuscript, carefully.

Third, authors suggest “We identified all patients who underwent digital subtraction angiography of the lower extremity arteries between 01/2020 and 11/2024.” (L74), but it is difficult to understand how to gather and select participants of this study and what the characteristics of participants (e.g., nationality and range of age). Authors should add flow chart of selection of participants and explain how to select them as well as their characteristics in method section.

Fourth, authors suggest “A normal forefoot PVR was defined by a sharp upstroke, a scooped or flat interval between peaks and a dicrotic notch at the diastolic descending part of the curve. PVR was classified as mildly abnormal when the dicrotic notch and the flat interval between curves were lost, while the upstroke remained steep. PVR was classified as severely abnormal when in addition to the above mentioned changes the amplitude was flattened and the upstroke got prolonged (equal upstroke and downslope period).” (L109), and “after revascularization” (L120), but they do not explain them using quantitative definition. Without quantity indicators, it is difficult to identify them, scientifically. Authors should add quantitative definition in method section.

Fifth, authors suggest “seventy patients were eligible for analysis” (L159) and “allowing limb-based analysis in a total of 90 cases” (L176), but this means violation of assumption regarding independency for statistical analyses (e.g., SD). Authors should use adequate statistical analysis (e.g., mixed analysis).

Sixth, authors used “Clinical symptoms and signs of infection” (Table 2), “Infection was clinically evident in almost two thirds of cases” (L184) and “Wound swabs were positive in 48 cases” (L186), but they do not explain the definition. It is difficult to understand what authors did, without their definitions. Authors should add the definitions in method section.

Seventh, authors used Figure 2, but it is difficult to understand what variation in each color means. Authors should add explanation regarding Figure 2.

Eighth, authors did not show result of samples in P2 in Table 3 without explanation, and they used only 26 samples. Moreover, the number of samples varies through result section including subgroup analyses, and it is difficult to understand what authors explain in result section. Authors should add tables as well as flow chart to explain difference in the number of samples and participants in each analysis, and revise the result section, carefully.

Nineth, authors suggest “An abnormal PVR curve morphology (mildly or severely abnormal) yielded” (L239), but it is difficult to understand why they do not include participants with non-pulsatile. Moreover, authors suggest “”, 

Tenth, authors suggest “When analyzing the diagnostic accuracy of the variables for detection of any PAD (GLASS > 1) or for severe PAD (GLASS > 2), ROC analysis revealed the highest area under the curve (AUC) for MSA (AUC 0.89 and 0.83). UST (AUC 0.82 and 0.71) and USR (AUC 0.81 and 0.76) both had worse diagnostic accuracy. With a cut-off value of 0.58 mmHg, MSA had a sensitivity of 91.4% and a specificity of 80.8% for detection of any PAD (GLASS > 1). For detection of severe PAD (GLASS > 2), MSA with a cutoff of 0.27 mmHg had a sensitivity of 72.2% and a specificity of 77.1% (Figure 5).” (L261), but they do not show 95%confidence intervals. Authors should show 95% confidence intervals for AUC, sensitivity and specificity.

Eleventh, authors suggest “Mean values of UST and USR did not differ between inframalleolar descriptor categories” (L271), “Within this subgroup, 19 cases had no significant PAD, whereas 15, 17 and 3 cases had GLASS stage I, III, and III, respectively on angiography.” (L295), and “No significant differences between patients with and without improved foot perfusion after revascularization were found for UST and USR.” (L321), but they do not show results of statistical analyses. Authors should add results of statistical analysis in result section as well as tables.

Twelfth, authors suggest “Informed Consent Statement: Patient consent was waived due to a retrospective cohort study design.” (L390), but this explanation is incorrect, because “retrospective cohort study design” is not always considered to be waived. Authors add the explanation by the Institutional Review Board why they allowed them to waive informed consent.

Finally, authors described some of sentences without citation or justification as follows; “Diabetic foot syndrome (DFS) is an important, frequent complication of diabetes mellitus, affecting approximately 550 million persons living with diabetes mellitus worldwide.” (L45), “noninvasive diagnosis of PAD, which commonly affects the cruropedal arteries in diabetic patients, is impaired secondary to medial arterial calcification (MAC).” (L57), “PVR are easily obtained by assistant staff and semiquantitative analysis is simple. However, the evidence on diagnostic accuracy of PVR for the detection of PAD in diabetic and nondiabetic subjects exhibiting MAC is poor.” (L65), “the AngETM system (SOT Medical Systems, Maria Rain, Austria).” (L93), “The standard pressure applied to the forefoot cuffs was 50 mmHg.” (L97), “the below the ankle outflow was scored (inframalleolar descriptor: P0, at least one patent foot supplying artery with patent pedal arch; P1 severely diseased or absent pedal arch; P2 no artery crossing the ankle towards the foot). Based on the three arterial levels, a global staging of disease severity was determined (grades 0-III).” (L138), “KDIGO categories G3-5” (L168), “PAD is common in patients suffering from DFS (up to 50%).” (L335), “Angiography as an invasive procedure with arterial puncture carries a not negligible complication risk.” (L339), and “This is mainly related to the high prevalence of MAC of the below-the-knee and below the ankle-arteries in diabetic patients, making systolic ankle pressure measurements and ABI unreliable for the diagnosis of PAD.” (L341), but it is difficult for readers to judge them without references as evidence for each description. Authors should add references for these descriptions.

Minor comments 

L30: “Interobserver agreement was substantial” may be “Interobserver agreement was substantial”.

L38: “MSA outperformed UST and USR but showed…” may be “On the other hand, MSA outperformed UST and USR showed…”.

L385: “Funding: This research received no external funding.” should not in conclusion.

Comments on the Quality of English Language

Authors suggest “Interobserver agreement was substantial” (L30), “for detection of inframalleolar disease” (L35), and “An impaired inframalleolar circulation (inframalleolar GLASS descriptor P1 or P2) was present in 28.6%, 50%, 51.7% and 83.3%” (L242), “Interobserver agreement of PVR was tested in 36 cases” (L246), but it is difficult to understand what authors explain. 

L30: “Interobserver agreement was substantial” may be “Interobserver agreement was substantial”.

L38: “MSA outperformed UST and USR but showed…” may be “On the other hand, MSA outperformed UST and USR showed…”.

Author Response

3.1. First, authors suggest “There were significant differences in the intervention duration, the dose of the intervention, and the regions of the intervention compared with the control group.” (L205), but they do not describe statistical results. Without statistical results, it is difficult to understand what they explain. Authors should revise result session, carefully.

The mentioned sentence is not from our manuscript!

3.2. Second, authors suggest “Interobserver agreement was substantial” (L30), “for detection of inframalleolar disease” (L35), and “An impaired inframalleolar circulation (inframalleolar GLASS descriptor P1 or P2) was present in 28.6%, 50%, 51.7% and 83.3%” (L242), “Interobserver agreement of PVR was tested in 36 cases” (L246), but it is difficult to understand what authors explain. Authors should revise the manuscript, carefully.

We reorganized the methods section in order to improve readability and comprehensibility of our manuscript.

3.3. Third, authors suggest “We identified all patients who underwent digital subtraction angiography of the lower extremity arteries between 01/2020 and 11/2024.” (L74), but it is difficult to understand how to gather and select participants of this study and what the characteristics of participants (e.g., nationality and range of age). Authors should add flow chart of selection of participants and explain how to select them as well as their characteristics in method section.

We added a flowchart depicting patient selection (please see also our response to comments 1.2. and 2.3.)

3.4. Fourth, authors suggest “A normal forefoot PVR was defined by a sharp upstroke, a scooped or flat interval between peaks and a dicrotic notch at the diastolic descending part of the curve. PVR was classified as mildly abnormal when the dicrotic notch and the flat interval between curves were lost, while the upstroke remained steep. PVR was classified as severely abnormal when in addition to the above mentioned changes the amplitude was flattened and the upstroke got prolonged (equal upstroke and downslope period).” (L109), and “after revascularization” (L120), but they do not explain them using quantitative definition. Without quantity indicators, it is difficult to identify them, scientifically. Authors should add quantitative definition in method section.

In this point we disagree. We described in detail how we quantified PVR (amplitude, upstroke time and upstroke ratio), in addition to the semiquantitative analysis mentioned by the reviewer. Most of our analyses rely on quantitative measurements and as a main result of our study we provide reference values for the quanitative parameters.

3.5 Fifth, authors suggest “seventy patients were eligible for analysis” (L159) and “allowing limb-based analysis in a total of 90 cases” (L176), but this means violation of assumption regarding independency for statistical analyses (e.g., SD). Authors should use adequate statistical analysis (e.g., mixed analysis).

With regard to this comment, we sought statistical advice (Dr. Crispin, Institute for Biometrics and Medical Statistics, Ludwig-Maximilians-University, Munich, Germany). He attested us that the statistical methods we applied are appropriate to answer our study questions and scientifically sound. It can of course be assumed that measurements on two legs of the same patient provide dependent, correlated results, but this should not have affected the assessment of the diagnostic test accuracy for detection of PAD in one leg at one time point (limb based analysis). Instead of performing additional statistics, he advised us to perform a sensitivity analysis including only the first examination per patient to enhance the robustness of our analysis. The results of this analysis were included in the revised manuscript (see also response to 1.3.).

3.6. Sixth, authors used “Clinical symptoms and signs of infection” (Table 2), “Infection was clinically evident in almost two thirds of cases” (L184) and “Wound swabs were positive in 48 cases” (L186), but they do not explain the definition. It is difficult to understand what authors did, without their definitions. Authors should add the definitions in method section.

We agree and provided the clinical definition of infection in the revised manuscript (methods section).

3.7. Seventh, authors used Figure 2, but it is difficult to understand what variation in each color means. Authors should add explanation regarding Figure 2.

All colored fields in the circle diagram are labelled with a letter and a number according to the Wagner-Armstrong classification. We revised Figure 2 in order to improve comprehensibility

3.8. Eighth, authors did not show result of samples in P2 in Table 3 without explanation, and they used only 26 samples. Moreover, the number of samples varies through result section including subgroup analyses, and it is difficult to understand what authors explain in result section. Authors should add tables as well as flow chart to explain difference in the number of samples and participants in each analysis, and revise the result section, carefully.

We agree that the text referring on table 3 should be improved, and revised the manuscript accordingly. In addition we revised table 3 to improve comprehensibility. The table covers the complete cohort (90 legs, six of whom with nonpulsatile PVR).

3.9. Nineth, authors suggest “An abnormal PVR curve morphology (mildly or severely abnormal) yielded” (L239), but it is difficult to understand why they do not include participants with non-pulsatile. Moreover, authors suggest “”, 

We addressed the small group of patients with nonpulsatile PVR in the next sentence: “A severely abnormal or non-pulsatile PVR yielded a sensitivity of 34.3% and a specificity of 96.6% for detection of very severe PAD (GLASS 3)”.

3.10. Tenth, authors suggest “When analyzing the diagnostic accuracy of the variables for detection of any PAD (GLASS > 1) or for severe PAD (GLASS > 2), ROC analysis revealed the highest area under the curve (AUC) for MSA (AUC 0.89 and 0.83). UST (AUC 0.82 [95% CI 0.73 – 0.91] and 0.71) and USR (AUC 0.81 []and 0.76) both had worse diagnostic accuracy. With a cut-off value of 0.58 mmHg, MSA had a sensitivity of 91.4% and a specificity of 80.8% for detection of any PAD (GLASS > 1). For detection of severe PAD (GLASS > 2), MSA with a cutoff of 0.27 mmHg had a sensitivity of 72.2% and a specificity of 77.1% (Figure 5).” (L261), but they do not show 95%confidence intervals. Authors should show 95% confidence intervals for AUC, sensitivity and specificity.

Thank you for this important advice. 95% CI for AUCs were added, as suggested.

3.11. Eleventh, authors suggest “Mean values of UST and USR did not differ between inframalleolar descriptor categories” (L271), “Within this subgroup, 19 cases had no significant PAD, whereas 15, 17 and 3 cases had GLASS stage I, III, and III, respectively on angiography.” (L295), and “No significant differences between patients with and without improved foot perfusion after revascularization were found for UST and USR.” (L321), but they do not show results of statistical analyses. Authors should add results of statistical analysis in result section as well as tables.

We agree and added the requested information for the sentences in L271 and L321. The information in sentence L295 is complete.

3.12. Twelfth, authors suggest “Informed Consent Statement: Patient consent was waived due to a retrospective cohort study design.” (L390), but this explanation is incorrect, because “retrospective cohort study design” is not always considered to be waived. Authors add the explanation by the Institutional Review Board why they allowed them to waive informed consent.

We clarified this issue, as requested.

3.13. Finally, authors described some of sentences without citation or justification as follows; “Diabetic foot syndrome (DFS) is an important, frequent complication of diabetes mellitus, affecting approximately 550 million persons living with diabetes mellitus worldwide.” (L45), “noninvasive diagnosis of PAD, which commonly affects the cruropedal arteries in diabetic patients, is impaired secondary to medial arterial calcification (MAC).” (L57), “PVR are easily obtained by assistant staff and semiquantitative analysis is simple. However, the evidence on diagnostic accuracy of PVR for the detection of PAD in diabetic and nondiabetic subjects exhibiting MAC is poor.” (L65), “the AngETM system (SOT Medical Systems, Maria Rain, Austria).” (L93), “The standard pressure applied to the forefoot cuffs was 50 mmHg.” (L97), “the below the ankle outflow was scored (inframalleolar descriptor: P0, at least one patent foot supplying artery with patent pedal arch; P1 severely diseased or absent pedal arch; P2 no artery crossing the ankle towards the foot). Based on the three arterial levels, a global staging of disease severity was determined (grades 0-III).” (L138), “KDIGO categories G3-5” (L168), “PAD is common in patients suffering from DFS (up to 50%).” (L335), “Angiography as an invasive procedure with arterial puncture carries a not negligible complication risk.” (L339), and “This is mainly related to the high prevalence of MAC of the below-the-knee and below the ankle-arteries in diabetic patients, making systolic ankle pressure measurements and ABI unreliable for the diagnosis of PAD.” (L341), but it is difficult for readers to judge them without references as evidence for each description. Authors should add references for these descriptions.

We appreciate this advice and added the respective references underlying our statements, where appropriate.

Minor comments 

3.14. L30: “Interobserver agreement was substantial” may be “Interobserver agreement was substantial”.

There is no difference between the written sentence and the suggested sentence.

3.15. L38: “MSA outperformed UST and USR but showed…” may be “On the other hand, MSA outperformed UST and USR showed…”.

We would favour the original sentence.

3.16. L385: “Funding: This research received no external funding.” should not in conclusion.

Agreed and corrected.

Comments on the Quality of English Language

3.17. Authors suggest “Interobserver agreement was substantial” (L30), “for detection of inframalleolar disease” (L35), and “An impaired inframalleolar circulation (inframalleolar GLASS descriptor P1 or P2) was present in 28.6%, 50%, 51.7% and 83.3%” (L242), “Interobserver agreement of PVR was tested in 36 cases” (L246), but it is difficult to understand what authors explain. 

The first, third and fourth sentence mentioned (L30, L242, L246) in our opinion are clear and we decided not to change these sentences. However, we revised the second sentence (L35).

3.18. L30: “Interobserver agreement was substantial” may be “Interobserver agreement was substantial”.

See 3.14.

3.19. L38: “MSA outperformed UST and USR but showed…” may be “On the other hand, MSA outperformed UST and USR showed…”.

See 3.15.

Reviewer 4 Report

Comments and Suggestions for Authors

This manuscript delves into the diagnostic effectiveness of pulse volume recordings (PVR) at the forefoot level in identifying peripheral arterial disease (PAD) in patients with diabetic foot syndrome (DFS). The authors conduct an analysis of PVR in conjunction with invasive angiography and various parameters including upstroke time (UST), upstroke ratio (USR), and maximum systolic amplitude (MSA) to ascertain the sensitivity and specificity of PVR in detecting PAD.

Strengths of the Manuscript

1. Novelty and Contribution: The manuscript provides a unique and focused exploration of PVR in diagnosing PAD, demonstrating its potential diagnostic utility in DFS. This addresses a gap in existing literature, making a significant contribution to this specialized field and laying the foundation for future research.

2. Article Structure and Writing: The manuscript is nicely and clearly written, with its claims mostly supported by scientific data.

Areas for Improvement

1. Introduction Section: The brief statement about the ease of obtaining PVR and simple semiquantitative analysis is insufficient to fully justify the use of PVR for PAD diagnosis. A more detailed explanation is needed, along with comparing PVR with other noninvasive diagnostic methods in the introduction rather than the discussion section.

2. Parameter Justifications: Provide reasoning for selecting parameters like the applied pressure of 50 mmHg, time resolution of 1ms, and amplitude resolution of 18 bits.

3. Discussion Section: Include a thorough analysis or speculation on why PVR shows diagnostic efficacy for PAD.

4. Grammar: Correct a minor grammatical error in line 64 by changing "have" to "has" in the phrase "transcutaneous oxygen measurement has been proposed."

Author Response

4.1. Introduction Section: The brief statement about the ease of obtaining PVR and simple semiquantitative analysis is insufficient to fully justify the use of PVR for PAD diagnosis. A more detailed explanation is needed, along with comparing PVR with other noninvasive diagnostic methods in the introduction rather than the discussion section.

We agree and extended our discussion, accordingly

4.2. Parameter Justifications: Provide reasoning for selecting parameters like the applied pressure of 50 mmHg, time resolution of 1ms, and amplitude resolution of 18 bits.

Done (parameters were used as specified by the manufacturer).

4.3. Discussion Section: Include a thorough analysis or speculation on why PVR shows diagnostic efficacy for PAD.

To properly address this comment, we broadened the discussion substantially (see also 1.1.)

4.4. Grammar: Correct a minor grammatical error in line 64 by changing "have" to "has" in the phrase "transcutaneous oxygen measurement has been proposed.”

The full sentence starts with: “Several alternative noninvasive diagnostic measures, including systolic toe pressure measurement and transcutaneous oxygen measurement have been proposed…”. Therefore, “have” is the correct form.

Reviewer 5 Report

Comments and Suggestions for Authors

The manuscript “Segmental pulse volume recordings at the forefoot level as a useful diagnostic tool for detection of peripheral arterial disease in the diabetic foot syndrome “ by Nützel et al. reports the analysis of the diagnostic yield of PVR of the forefoot for detection of PAD of limbs affected by DFS. This work is well written, but did authors ever try to separate the normal and moderately impaired results? Therefore, I would suggest authors may take at least a revision. Here are the comments and suggestions:

1.      There are some typos throughout this work.

2.      In Table 3, the results of P2 can be added.

3.      In Figs. 3-5, should authors also separate P0 and P1, and add P2?

4.      The conclusions can be extended, and the Funding can be separated.

Author Response

5.1.      There are some typos throughout this work.

We carefully revised the manuscript with regard to typos.

5.2.  In Table 3, the results of P2 can be added.

Done (see also 3.8.)

5.3.  In Figs. 3-5, should authors also separate P0 and P1, and add P2?

In order to preserve clarity, we decided against inclusion of the inframalleolar descriptor in these figures.

5.4. The conclusions can be extended, and the Funding can be separated.

We extended the conclusions (see also 1.4. and 3.16.)